# Dysregulation of FURIN and Other Proprotein Convertase Genes in the Progression from HPV Infection to Cancer

**DOI:** 10.3390/ijms26020461

**Published:** 2025-01-08

**Authors:** Gonzalo Izaguirre, Natalia Zirou, Craig Meyers

**Affiliations:** 1Department of Periodontics, College of Dentistry, University of Illinois Chicago, Chicago, IL 60612, USA; 2Departments of Microbiology and Immunology, College of Medicine, Penn State University, Hershey, PA 17033, USA

**Keywords:** HPV, FURIN, proprotein convertases, cancer, bioinformatics

## Abstract

Productive infections of oncogenic human papillomaviruses (HPVs) are closely linked to the differentiation of host epithelial cells, a process that the virus manipulates to create conditions favorable to produce virion progeny. This viral interference involves altering the expression of numerous host genes. Among these, proprotein convertases (PCs) have emerged as potential oncogenes due to their central role in cellular functions. Using RT-qPCR, aberrant PC gene expression was detected across the progression from early HPV infection stages to cancer. These findings demonstrated a progressive disruption of normal PC expression profiles, with FURIN consistently downregulated and other PCs upregulated, transitioning from the episomal stage to neoplastic and carcinoma phenotypes. This pattern of dysregulation was distinct from the broader trends observed in a variety of cancer types through bioinformatic analysis of publicly available transcriptomic datasets, where FURIN expression was predominantly upregulated compared to other PCs. Further bioinformatic investigations revealed a correlation between PC gene expression and cancer phenotype diversity, suggesting a potential link between the loss of normal PC gene expression patterns and the progression of HPV infections toward malignancy.

## 1. Introduction

Proprotein convertases (PCs) of the constitutive protein secretion pathway, including FURIN, PC4, PC5, PACE4, and PC7, are key regulators of cellular metabolism. These enzymes perform post-translational proteolytic processing of precursor proteins, which are critical for central cellular functions such as growth, proliferation, and differentiation [1]. As serine proteases of the subtilase family, PCs cleave substrates at polybasic sites, which are minimally composed of two arginine residues at positions P4 and P1 (P4R-X-X-P1R ^ P1′, where cleavage occurs between P1 and P1′). PCs exist in membrane-bound and free forms and localize within the trans-Golgi network, endosomes, and the pericellular space.

The ubiquitous presence of PCs in mammalian cells has enabled many viruses to exploit these proteases to facilitate their infective life cycles [2]. For example, HPV relies on PCs for the proteolytic activation of its capsid proteins, which is essential for infection competence [3,4,5]. Unlike many viruses, HPV’s life cycle is tightly integrated with the differentiation program of its target cells—infected basal keratinocytes that eventually transform into virus-releasing squamous cells [6]. Even in the early stages of infection, HPV induces significant changes in host gene expression, modulating cellular metabolism to favor virus progeny production while evading immune detection [7,8,9]. During the productive infection stage, the viral chromosome replicates as an episome; however, chronic infection can lead to the integration of the viral genome into the host’s chromosomal DNA. This integration halts virion production and induces chromosomal instability that may ultimately result in cancer.

PCs cleave a wide variety of substrates, including growth factor receptors and their ligands, adhesion molecules, and extracellular matrix components and regulators [10]. Many of these substrates are known to contribute to cancer development and maintenance, highlighting the role of PCs in oncogenesis [11]. Studies suggest that FURIN overexpression plays an oncogenic role due to its activation of tumorigenic substrates [12,13]. However, in other cancers, FURIN may act as a tumor suppressor. Similarly, other PCs have shown roles in cancer that sometimes oppose FURIN’s function within the same cancer type [14]. Therefore, the relative expression of individual PCs may be as important as their absolute levels of expression.

Given the significant changes in host gene expression during HPV-dependent oncogenic progression, this study analyzed the expression of PCs as a group and focused on transcriptional changes in keratinocytes at early and late stages of HPV infection and cell transformation using RT-qPCR. The results showed a clear progressive loss of the normal pattern of PC gene expression. These findings were further compared to PC gene expression in other normal and cancerous cell types through bioinformatic analyses of publicly available transcriptomic datasets. While FURIN is consistently overexpressed in most cancer types, our study with epithelial cells showed the opposite pattern, in which FURIN was consistently downregulated. Further bioinformatics analysis correlated changes in FURIN expression with the regulation of genes associated with HPV infection and various cancer types. These results suggest that the loss of the normal pattern of PC gene expression profile during HPV infection is closely linked to the cancer phenotype, marking a potential step in the progression toward malignancy.

## 2. Results

### 2.1. PC Gene Expression in Normal Human Keratinocytes Differs Between Organotypic and Monolayer Cultures

Primary keratinocytes were isolated from five anatomic sites of HPV infection and analyzed for mRNA expression by RT-qPCR. The PC gene expression counts were standardized to those of the GAPDH gene. The PC gene expression in organotypic rafts was variable among anatomic sites, but the pattern of individual gene expression was constant and dominated by FURIN, PC4, and PACE4, followed by PC7 and PC5 (Figure 1A). However, the same cells growing in monolayer cultures expressed PC genes about 20-fold lower and in a different pattern dominated by FURIN, followed by PC7, PACE4, PC4, and then by PC5 (Figure 1B).

### 2.2. HPV Infection Affects PC Gene Expression in Human Keratinocytes

The same primary keratinocytes described above growing in monolayer cultures were electroporated with the HPV16 circular chromosome for the episomal expression of viral genes. The viral genome affected PC gene expression when compared to non-HPV-infected cells (Figure 2A). FURIN expression became downregulated in some anatomic sites. The integration of the viral genome into the host cell’s genome is a further step into the infection process and cancer progression. Several cell lines derived from clinical HPV-positive neoplasia and carcinoma samples harboring multiple integrated copies of the viral genome were tested for their level of PC gene expression (Figure 2B). In the three neoplasia-derived cells, PC gene expression was also affected, and FURIN was downregulated in all of them. In the three carcinoma derived cells, PC gene expression was affected to a larger extent compared to the neoplasia cells, with FURIN being mostly downregulated.

### 2.3. HPV-Negative Epithelial Cancer Cell Lines Lost the Normal PC Gene Expression Pattern

The PC gene expression in several established HPV-negative epithelial cancer cell lines was compared to that in the non-HPV infected control cells shown in Figure 1B. The immortalized N-TERT cell line was practically no different from the control (Figure 3). These cancer cell lines showed PC gene expression profiles largely more affected compared to the HPV-positive carcinomas shown above, and in all of them, FURIN was downregulated. Taken together, the analysis of episomal HPV-infected cells, HPV-positive neoplasia and carcinoma cells, and HPV-negative cancer cells showed not only a progressive loss of the normal pattern of PC gene expression observed in normal cells, but also an increased variability in gene expression, with the FURIN gene being downregulated (Figure 4).

### 2.4. Bioinformatic Analysis of Large Transcriptomic Datasets—Normal Cells

To acquire an overview of the pattern of PC gene expression in a wider range of cell types, we accessed publicly available transcriptome datasets, including a large collection of tissues and cell lines with normal or cancer phenotypes, and performed bioinformatic analyses. The first dataset consisted of 50 normal tissues, and the gene expression values were extracted and averaged for each PC gene (Figure 5). The data were also analyzed to obtain the expression count modal bins, modal frequencies, and lowest and highest range values for the five PC genes and the GAPDH control gene across all tissues (Appendix A, Appendix A). A similar analysis was performed on a dataset containing the transcriptome of single-cell suspensions from 81 normal tissues without pre-enrichment of cell types (Figure 5). For the latter dataset, range values were calculated by averaging the 2–5 most extreme values at each end (Appendix A, Appendix A). The averaged single-cell data were more variable; however, the same pattern with PC7 followed by FURIN as the genes with the largest modal expression values was observed across normal tissues and single-cell samples. However, although this was the predominant pattern, individual normal cell types may show a different modal expression pattern.

### 2.5. PC Gene Expression in Cancer Samples and Cell Lines

An additional dataset included the transcriptome of 7932 samples belonging to 21 different cancer types (Appendix A). The gene expression values were averaged for each PC gene (Figure 5). Modal bins and their frequencies, and lowest and highest range values of PC gene expression were determined (Appendix A, Appendix A). FURIN produced the highest average and modal expression values compared to the other PCs, whose expression values were similar. A dataset containing the transcriptome of 1206 cancer cell lines was also analyzed (Figure 5 and Appendix A, Appendix A). FURIN had the largest average expression value, but it was not as predominant in the cell lines as with the cancer samples. The cancer cell lines had FURIN and PC7 with the highest modal expression values, noticeably differing from the predominance of only FURIN in the cancer tissue samples.

### 2.6. Cancer Correlates with Dysregulation of PC Gene Expression

To compare the PC gene expression between datasets of normal and cancer cells, the mean values in Figure 5 were standardized to the GAPDH gene expression and multiplied by ten thousand to avoid fractional numbers (Appendix A). However, the large dispersion of the data made it necessary to complement this analysis with an additional analysis based on the modal expression values. The modal distribution data were also standardized to the expression of the GAPDH gene by factoring the center value of the modal range, firstly by the ratio of the modal frequency over the total number of samples in the dataset, and secondly by the middle value of the expression range. The resultant value for each PC gene was then divided by the value that was similarly obtained for the expression of the GAPDH gene. These relative PC gene expression values were then multiplied by one million to avoid fractional numbers (Appendix A). Mean and modal derived relative expression values were converted into a percentage of their corresponding total PC gene expression value and averaged (Figure 6). The mean and modal percentage values for the normal tissues closely agreed (Appendix A). A major result concluded that FURIN expression in cancer samples dramatically increased compared to the normal tissues. FURIN was also the predominant PC expressed in cancer cell lines. The expression of FURIN was lower in normal single cells compared to normal tissues; otherwise, the expression pattern was similar. Together, these results show upregulation of FURIN with respect to the other PC genes in cancer, with the resultant loss of the PC gene expression pattern observed in normal cells.

### 2.7. FURIN Gene Expression Determines Different Cancer Phenotypes

The above bioinformatic analyses showed that the expression of FURIN becomes dominant over the other PCs in cancer compared to the normal phenotypic state. We made additional comparisons of the global gene expression between two groups of six cancer cell lines each, one group with the lowest and the other with the highest levels of one PC gene expression. The cancer cell lines included in these analyses and their PC gene counts are listed in Appendix A. These multidimensional comparisons showed defined phenotypic difference, especially for FURIN (Figure 7), whose six lowest expressing cell lines grouped very close together, suggesting a very similar phenotype. In contrast, the six highest FURIN-expressing cell lines were more phenotypically variable. The cell lines expressing the extreme levels for PC5 and PACE4 grouped into distinctive but widely distributed phenotypes. In contrast, for PC7 and, especially, PC4, the cell line groups did not distribute homogeneously but more dispersedly. Altogether, these comparisons showed that the level of expression of some PC genes can be associated with different and distinctive cancer phenotypes. Further analyses of the gene expression differences between the lowest and highest FURIN-expressing cancer cell lines yielded that out of the 12,650 genes analyzed, 1646 were downregulated and 2992 upregulated in the low-FURIN-expressing cells relative to the high-expressing cells (*q* < 0.05), with FURIN being the gene with the largest difference (Figure 8 and Appendix A). A heatmap plot of the differential expression of the top 50 genes with the largest differences in expression between the lowest and highest FURIN-expressing groups of cells clearly showed the stark differences between these two cancer phenotypes.

### 2.8. HPV-Infection-Related Genes Are Differentially Expressed Between High and Low FURIN Expressing Cancer Cells

The genes (4584) that showed significant difference in expression between the lowest and highest FURIN-gene-expressing cell groups were analyzed for their ontological functional associations. Groups of genes were found associated with components of cell signaling, surface receptor, growth factor, and extracellular matrix functions (Table 1). Enrichment analysis within these functional association groups further identified categories and subcategories of human diseases related to smaller sets of genes (Appendix A). One association was of particular interest, and that was with a set of 64 genes involved in HPV infection (Appendix A). Further enrichment to identify the pathways in which this set of 64 genes participates showed an association with several forms of cancer (Figure 9).

## 3. Discussion

PCs play ubiquitous and essential roles in regulating cell metabolism. Evidence of their dysregulation in cancer cells, particularly the overexpression of FURIN, is well documented, leading to proposals for anticancer therapeutic strategies targeting this enzyme [15,16]. However, FURIN may be overexpressed or underexpressed, depending on the cancer type. The relative expression of PC genes seems most relevant, as suggested by the opposite role observed between FURIN and other PCs in some cancer phenotypes [14]. This study addressed the changes in gene expression of FURIN and the other PCs during HPV-induced malignant transformation of keratinocytes, beginning at the initial stages of oncogenic HPV infections, advancing through epithelial HPV-positive neoplasia cell lines, and culminating in cancer cells. These changes were compared to the PC gene expression in several cancer and normal cells by bioinformatic analyses. Overall, these two approaches agreed that besides changes to FURIN expression, the combined change to all PC gene expression and the resultant loss of the normal PC gene expression pattern is what seems to characterize the PC dysregulation in the cancer phenotype.

The study highlights a 20-fold higher PC gene expression in keratinocytes cultured in organotypic rafts compared to monolayer cultures. This observation underscores the critical role of PCs in organogenesis and embryonic development, suggesting that higher PC levels support cell differentiation, while lower levels favor proliferation. Supporting this notion, normal cells grown in monolayers and cancer cells exhibit significantly reduced PC gene expression. Interestingly, the PC expression profile remains consistent across keratinocytes from the five anatomical sites of HPV infection, despite minor differences in total expression levels. However, keratinocytes grown in monolayers displayed a distinct PC expression profile different to that in raft cultures. The dysregulation of PC gene expression in HPV-infected cells suggests that it may be part of the mechanism that derails the cell differentiation process and switches to proliferation.

As HPV progresses from episomal to chromosomal genome insertion, there is a notable divergence in PC gene expression profiles, with a tendency for the FURIN expression to decrease. This trend points to a gradual loss of PC expression integrity that correlates with the disruption of the keratinocyte differentiation state. In epithelial cancer cell lines, PC expression became even more variable, both in total levels and profile composition. There was a generalized loss of PC gene expression, with some cancer cell lines exhibiting the lowest PC expression levels observed. Overall, FURIN was especially affected showing a consistent decrease in expression. This observation seems to contradict studies showing increasing presence of FURIN in cervical epithelial cancers [17]. Therefore, other cells should be responsible for producing FURIN in the tumor environment.

The analysis of large publicly available transcriptomic datasets revealed marked differences in PC gene expression patterns between normal and cancer tissues, with FURIN expression significantly elevated in cancer tissues. This finding aligns with studies reporting FURIN overexpression across various cancer types [12,13]. However, it is important to note that these findings reflect average trends across diverse cancer types and do not exclude the possibility of divergent PC expression profiles in specific cancers, as observed in keratinocyte-derived cell lines in this study. It should be noticed that epithelial HPV-positive cancers tend to respond better to treatment than the HPV-negative ones [18], highlighting the differences between these two types of cancer.

Further comparisons of the transcriptomes of cancer cell lines with the lowest and highest PC expression levels revealed distinct phenotypic differences. Notably, genes differentially expressed in low- and high-FURIN-expressing cancer cells included a subset of genes affected during HPV infections. This overlap suggests a potential link between the phenotypes of HPV infection and cancer development. These findings point to a probable role for PCs in cancer development during the progression of HPV infections, highlighting their potential significance in oncogenesis. The low FURIN expression status in the HPV-positive cancer indicates that the other PCs may be involved in sustaining this cancer phenotype. This premise suggests that PCs may be specific for different signaling pathways, and identifying these associations may be needed to better understand the molecular basis for the mechanism of HPV-dependent oncogenic progression.

## 4. Materials and Methods

### 4.1. Cell Lines and Cultures

Primary human keratinocytes (HK) were isolated from cervical biopsy specimens as previously described [8]. HK cell lines persistently infected with HPV16 (HK-HPV) were produced by electroporating primary HK with HPV16 chromosomal DNA. The electroporated cells were cultured with mitomycin C-treated J2 3T3 feeder cells. Both normal and HPV16-immortalized HFK, HCK, HVK, HAK, HGK, and HTLK lines stably maintained episomal HPV16 DNA. Organotypic raft cultures were grown at the first or second passage for primary HK. The raft tissues were harvested after 20 days for RT-qPCR. Cancer cell lines were purchased from the ATCC and cultured as described by the vendor.

### 4.2. Measurement of Gene Expression by RT-qPCR

The PC gene expression in primary keratinocytes was measured from mRNA that was isolated using RNeasy kit (Qiagen, Germantown, MD, USA), and cDNA synthesis was performed using the Maxima First Stranded kit (Thermo-Fisher, Waltham, MA, USA). Quantification of transcriptional gene expression by RT-qPCR was performed using the standard SYBR Select Master Mix (Applied Biosystems, Waltham, MA, USA, Life Technologies, Carlsbad, CA, USA) in a Bio-Rad CFX96 Real-Time System thermal cycler programmed for 40 cycles. Expression values were the result of two independent determinations with three repeats each. Gene expression values were presented as their average mean and standard deviation. In Figure 1A, an independent two-tailed Student *t*-test was used to estimate differences among the groups composed of each of the expression values of the five PCs. The corresponding forward and reverse primers based on the human sequences for each gene were obtained from The Primer Bank and synthesized by Integrated DNA Technologies, and those were, for FURIN (ID 20336193c2), 5′-tcggggactattaccacttctg-3′ and 5′-ccagccactgtacttgaggc-3′; for PC4 (ID 20336189c9), 5′-gctgccggtcggaaatgaa-3′ and 5′-gtcgtagctggcgtaggaat-3′; for PC5 (ID 207030317c2), 5′-gagggacccacagtttcatttc-3′ and 5′-tgggcacgactgaagtcataa-3′; for PACE4 (ID 20336189c2), 5′-gctgccggtcggaaatgaa-3′ and 5′-gtcgtagctggcgtaggaat; for PC7 (ID 20336247c1), 5′-gcagcgtccacttcaacga-3′ and 5′-gcccagtcacattgcgttc-3′; and for GAPDH (ID 378404907c1), 5′-ggagcgagatccctccaaaat-3′ and 5′-ggctgttgtcatacttctcatgg-3′.

### 4.3. Bioinformatics Analysis

The sources of gene expression data from normal tissues and cells were the Human Protein Atlas (HPA—https://www.proteinatlas.org, accessed on 18 September 2024) [19] and the Genotype-Tissue Expression Project (GTEx—https://gtexportal.org, accessed on 19 September 2024) [20], for cancer samples was The Cancer Genome Atlas (TCGA—https://www.cancer.gov/tcga, accessed on 23 September 2024) [21], and for cancer cell lines were the HPA and the Cancer Cell Line Encyclopedia (CCLE—https://depmap.org/portal/ccle/, accessed on 23 September 2024) [22] through the portal of the Cancer Dependency Map (DepMap—https://depmap.org/portal/, accessed on 19 September 2024) [23]. The data analysis was performed using the software Rstudio 4.4.1 and several Bioconductor packages and libraries that included edgeR, readxl, GEOquery, bioaRt, dplyr, tidyr, writexl, pheatmap, ggplot2, clusterProfiler, KEGGREST, ReactomePA, org.Hs.eg.db, hsapiens_gene_ensembl, GO.db, and AnnotationDbi. Gene counts were downloaded, converted to text files, and uploaded into Rstudio [24,25]. The analysis workflow started with the filtering, normalization (TMM), and annotation of the data. Datasets were then converted into DGEList objects, followed by data exploration by multidimensional scaling (MDS), and dispersion estimation using the biological coefficient of variation (BCV). The differential gene expression analysis included the proper matrix design and fittings to a generalized linear model (Glms) using quasi-likelihood (QL) F-test to produce a list of significantly expressed genes and the fitting dispersion quantification. The differentially expressed genes were subjected to gene ontology and pathway analysis. The code lines for each of the analyses performed are included in the Appendix A.

## 5. Conclusions

PCs are essential regulators of the delicate balance between cellular differentiation and proliferation. Given the sensitivity of this mechanism, it is unsurprising that PCs play a pivotal role in pathological states such as cancer and viral infections. This study highlights PCs as a potential critical link in the progression of HPV infections toward malignancy, underscoring their significance in both viral pathogenesis and cancer development.

## Figures and Tables

**Figure 1 ijms-26-00461-f001:**
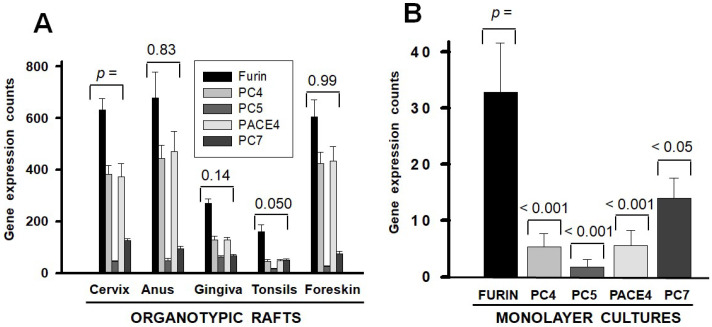
PC gene expression in non-HPV-infected human keratinocytes. Primary keratinocytes isolated from anatomic sites of HPV infection were grown in organotypic rafts (panel **A**), or in monolayer cultures (panel **B**). PC gene expression was determined by RT-qPCR and expression count values were standardized to the cDNA concentration and expression of the GAPDH gene. In panel (**B**), PC gene expression values represent the averages from the five anatomic sites. Data in panel (**A**) have been previously published but in a different format [5]. Statistical difference significance *p*-values are shown for the comparisons against the Cervix, in panel (**A**), and against the FURIN gene, in panel (**B**), using two-tailed Student *t*-test. Gene expression values represent means and standard deviations (error bars). Plots were made with SigmaPlot 14.5 software.

**Figure 2 ijms-26-00461-f002:**
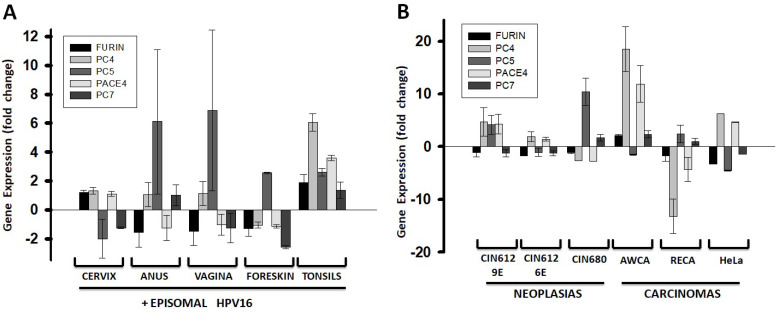
Fold changes of PC gene expression in HPV-infected human keratinocytes. In panel (**A**), primary keratinocytes isolated from anatomic sites of HPV infection and grown in a monolayer were electroporated with the HPV16 chromosome, which replicates as an episome. In panel (**B**), cell lines derived from clinical samples of HPV-positive neoplasia (CIN612, CIN680) or carcinoma (AWCA, RECA, HeLa) harboring several copies of the HPV31 (CIN612), HPV16 (CIN680), or HPV18 (AWCA, RECA, HeLa) genome. PC gene expression was determined by RT-qPCR as described in the legend to Figure 1. Gene expression values are relative to the PC gene expression in normal keratinocytes (Figure 1B) and represent means and standard deviations (error bars). Plots were made with SigmaPlot software.

**Figure 3 ijms-26-00461-f003:**
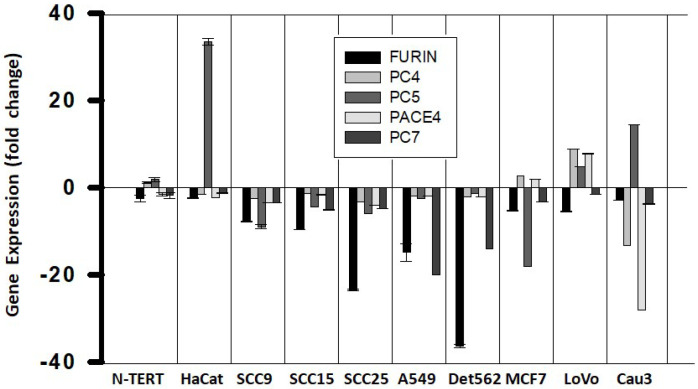
Fold changes of PC gene expression in HPV-negative epithelial cancer cell lines. Cell lines were obtained from the ATCC. PC gene expression was measured as described in the legend to Figure 1. Gene expression values are relative to the PC gene expression in normal keratinocytes (Figure 1B) and represent means and standard deviations (error bars). Plots were made with SigmaPlot software.

**Figure 4 ijms-26-00461-f004:**
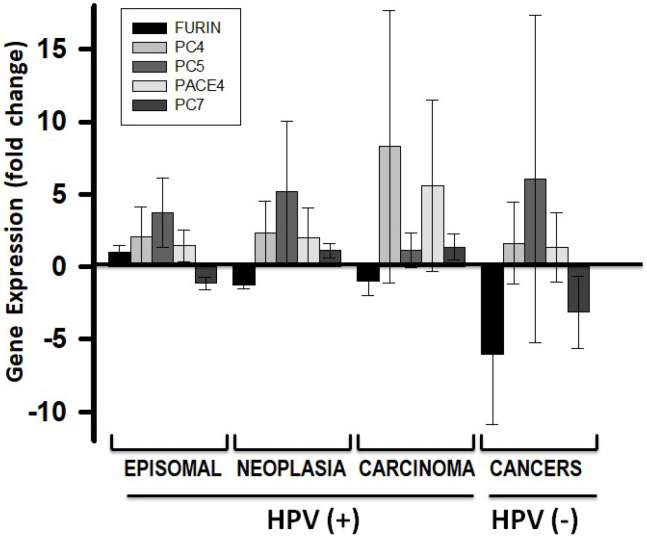
Progression of PC gene expression dysregulation in HPV-positive cancers. Gene expression values were averaged for each progression stage group and presented as fold changes with respect to non-HPV-infected keratinocytes. Plotted values represent the mean and standard deviation (error bars). The plot was made with SigmaPlot software.

**Figure 5 ijms-26-00461-f005:**
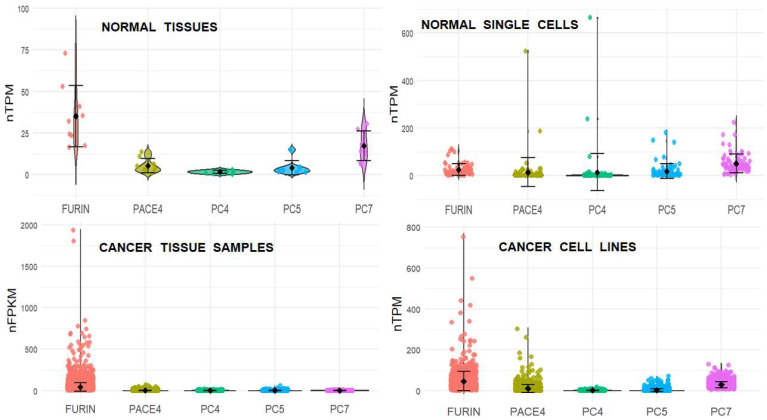
Comparison of PC gene expression in normal and cancerous cells. Plots show distribution of expression values (color dots) along with the mean (black dots) and associated standard deviation (error bars) for each PC gene. Datasets each contain 50 (normal tissues), 81 (normal single cells), 7932 (cancer tissue samples), or 1206 (cancer cell lines) data points. Plots were produced using Rstudio software and the ggplot2 package.

**Figure 6 ijms-26-00461-f006:**
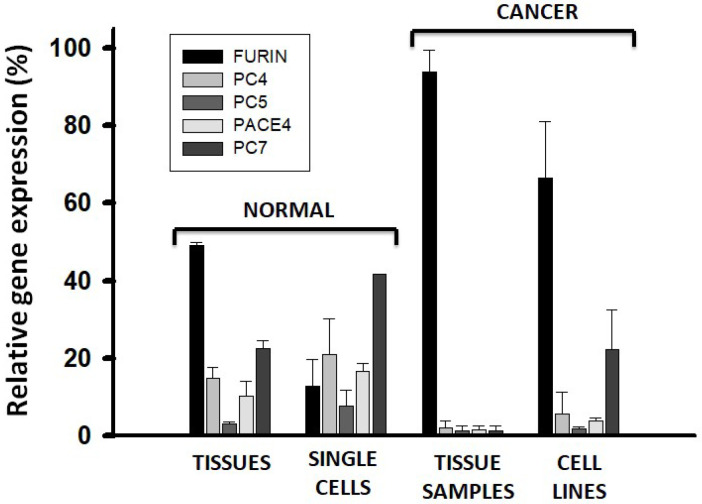
Dysregulation of PC gene expression in cancerous cells. Gene expression values obtained from combining mean and modal distribution analyses. Plots represent the average of the two values, and their associated range is represented by the error bars. The plot was made with SigmaPlot software.

**Figure 7 ijms-26-00461-f007:**
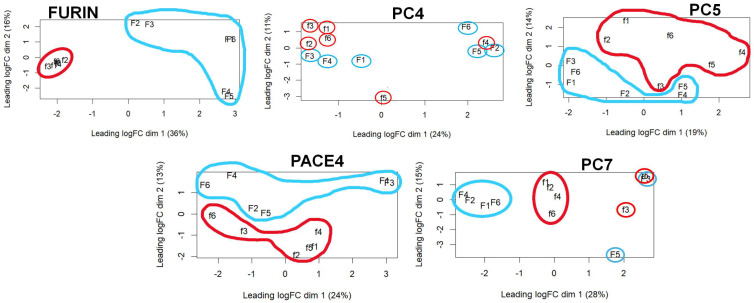
Multidimensional scaling comparison of cancer phenotypes based on PC gene expression differences. Each MDS plot shows the comparison between two groups of six cancer cell lines each. One group consisted of cells with the highest expression (F—blue) of a PC and the other group consisted of cells with the lowest (f—red). The identities of all the cell lines are listed in Appendix A. Plots were produced using the Rstudio software.

**Figure 8 ijms-26-00461-f008:**
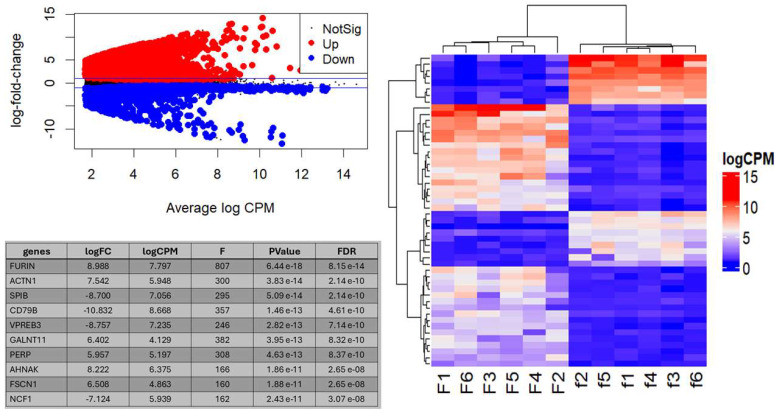
Differential gene expression analysis between the high- and low-FURIN-expressing cell lines. Upper left panel shows the MA plot of the average expression of genes (log2 counts per million) versus log2 fold-change in gene expression between conditions F and f. Red and blue dots represent genes that are significantly upregulated or downregulated, respectively. The right-side panel shows the hierarchical clustered heatmap of the 50 most differentially expressed genes between the F and f conditions. The table at the lower left panel shows the identity of the 10 most differentially expressed genes between the F and f cell line groups, the log2 of the fold change in their expression, the log2 average expression level of the gene across all samples, the statistical F value, the *p*-value, and the false discovery rate value. Plots were produced using Rstudio software and the ComplexHeatmap package.

**Figure 9 ijms-26-00461-f009:**
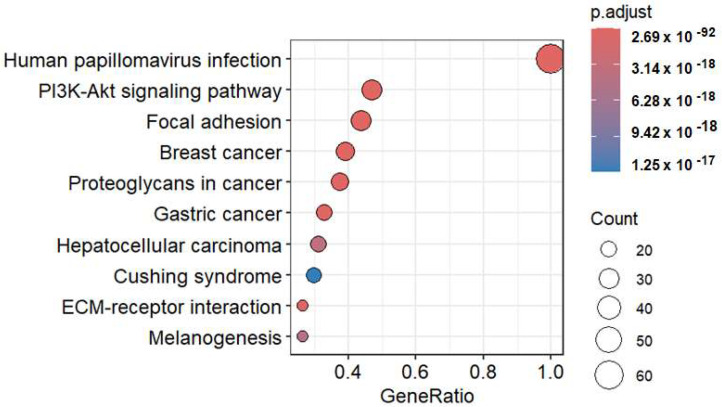
Dot plot showing the pathway analysis of a subgroup of genes differentially expressed between groups of cancer cell lines expressing the lowest and highest levels of the FURIN gene. The functional enrichment analysis of the genes that were differentially expressed between the FURIN F and f cell groups identified a set of 64 genes involved with the HPV infection. Further pathway analysis identified mechanisms and pathways potentially associated with this set of 64 genes. Plot was produced using the Rstudio software.

**Table 1 ijms-26-00461-t001:** Ontological functional associations of the genes differentially expressed in cancer cell lines between low and high levels of FURIN expression. N = number of genes associated with the term; CC = cellular component; BP = biological process; MF = molecular functions; Up = number of upregulated genes; Down = number of downregulated genes; P. Up and P. Down = *p* value for the corresponding change in gene expression.

Key	Term	Ont	N	Up	Down	P. Up	P. Down
GO: 0023052	Signaling	BP	4006	1224	421	7.29 × 10^−36^	9.99 × 10^−1^
GO: 0007166	Cell surface receptor signaling pathway	BP	1855	655	166	9.96 × 10^−36^	1.00 × 10^0^
GO:0005102	Signaling receptor binding	MF	883	372	78	4.72 × 10^−37^	9.99 × 10^−1^
GO:0038023	Signaling receptor activity	MF	493	221	49	2.49 × 10^−26^	9.78 × 10^−1^
GO:0019838	Growth factor binding	MF	98	69	1	8.55 × 10^−23^	9.99 × 10^−1^
GO:0030545	Signaling receptor regulator activity	MF	219	115	11	8.48 × 10^−21^	9.99 × 10^−1^
GO:0030546	Signaling receptor activator activity	MF	206	109	9	4.12 × 10^−20^	9.99 × 10^−1^
GO:0048018	Receptor ligand activity	MF	201	107	9	4.99 × 10^−20^	9.99 × 10^−1^
GO:0004888	Transmembrane signaling receptor activity	MF	347	158	37	6.84 × 10^−20^	8.98 × 10^−1^
GO:0031012	Extracellular matrix	CC	267	177	7	8.88 × 10^−51^	1.00 × 10^0^

## Data Availability

The data presented in this study are available on request from the corresponding author.

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
