# Peer review of "Dysregulation of FURIN and Other Proprotein Convertase Genes in the Progression from HPV Infection to Cancer"

_ijms, 2025, doi:10.3390/ijms26020461_

Round 1
Reviewer 1 Report
Comments and Suggestions for Authors
Papillomaviruses interact intimately with host cells, both responding to and manipulating many signaling, growth control, and differentiation pathways. PCs are important regulators of these pathways, so Izaguirre et al examine the expression levels of a group of PCs in a wide range of cell lines, both experimentally and using publicly available databases. Although the subject is of great potential interest, several aspects of this paper limit enthusiasm.
1. It is not clear what the hypothesis driving the research is. “PC levels vary across cell types and tissues” is not a strong hypothesis. The fact that levels of something are “altered” is not interesting (by itself). I hesitate to criticize work as “observational” because observations are critical to science, but in this case, the reader does not come away with an understanding of the authors’ point or conclusions. What is the reader supposed to make of all this? They are getting close toward the end when they identify genes and pathways regulated in common by furin and HPV. The work would be much stronger if organized around a clear question. Why do they think those changes occur? Even better, what experiment could they do to show why?
2. Contributing to the confusion about the point is the data presentation. Bar graphs and tables are probably not the most effective ways to visually convey the data. The bioinformatics data could be shown using violin plots, for example. Things that are meant to be compared should be placed side by side. The effect of HPV should be shown as a fold change over uninfected cells. The data in Fig 4a could be shown as a cloud of data points rather than an average +/- huge error bars. And so forth. As it stands, we are seeing a lot of numbers, but it is hard to discern what point the authors are making.
3. Some of the samples were from cells grown in organotypic culture and some were from monolayer cell lines. This should be clearly spelled out in the figure legends, and preferably on the graphs themselves.
4. The statistical analysis is strange. In most of the figures, the authors seem to be comparing one group of numbers as a block to another group as a block to determine p values rather than individual data points. They also seem to gloss over what seem to be potentially interesting differences between PCs from sample to sample. One would think that the differences between PC levels at different sites is the thing that the authors are actually looking for, but generally those differences are not generally emphasized.
5. Fig 1B: Is this the average of the 5 anatomical sites included in A? Wouldn’t it be better to normalize the values for each anatomical site to furin and then average the sites together? That would represent the difference in overall levels between the enzymes in a site-agnostic way.
6. Fig 2B: it should be indicated on the graph or in the legend which cell types are neoplasias and which are carcinomas.
7. Fig 4B: what are these numbers and where do they come from? The levels of furin tend to be high in all the other data in the paper, but not as high as in this figure.
8. Fig 5: the groupings look hand drawn and vary in style from panel to panel.
Author Response
Papillomaviruses interact intimately with host cells, both responding to and manipulating many signaling, growth control, and differentiation pathways. PCs are important regulators of these pathways, so Izaguirre et al examine the expression levels of a group of PCs in a wide range of cell lines, both experimentally and using publicly available databases. Although the subject is of great potential interest, several aspects of this paper limit enthusiasm.
- It is not clear what the hypothesis driving the research is. “PC levels vary across cell types and tissues” is not a strong hypothesis. The fact that levels of something are “altered” is not interesting (by itself). I hesitate to criticize work as “observational” because observations are critical to science, but in this case, the reader does not come away with an understanding of the authors’ point or conclusions. What is the reader supposed to make of all this? They are getting close toward the end when they identify genes and pathways regulated in common by furin and HPV. The work would be much stronger if organized around a clear question. Why do they think those changes occur? Even better, what experiment could they do to show why?
Response: I appreciate the observation because it was a concern from the beginning how to present data that is merely descriptive and not necessarily hypothesis driven. The reason to do this study was to address the seemingly contradictory information about the role of furin in cancer, and the absence in past studies of considering the PCs as a functional group in which the relative expression of individual PCs is as important as the absolute levels of expression. As was stated in the abstract: “The results revealed a progressive disruption of normal PC expression profiles, coinciding with the transition from episomal to integrated viral genomes and, ultimately, to the cancer phenotype”, New figures were made to better show the progression from a consistent pattern of PC gene expression in normal cells to a ‘complete loss’ (instead of altered) of the normal pattern accompanied by high variability. The bioinformatics study showed that the HPV-dependent oncogenesis is a deviation from the more general trend observed with a majority of cancers. We have added new text in the abstract, introduction, results and discussion sections to strengthen this line of thought. - Contributing to the confusion about the point is the data presentation. Bar graphs and tables are probably not the most effective ways to visually convey the data. The bioinformatics data could be shown using violin plots, for example. Things that are meant to be compared should be placed side by side. The effect of HPV should be shown as a fold change over uninfected cells. The data in Fig 4a could be shown as a cloud of data points rather than an average +/- huge error bars. And so forth. As it stands, we are seeing a lot of numbers, but it is hard to discern what point the authors are making.
Response: New figures were made that show the ratios relative to the gene expression in the non-HPV infected cells. One effect of HPV infection is an increase in variability of gene expression; therefore, the more maligned samples show higher variability. In the bioinformatics section, the tables were moved to the supplement and a new figure combining all the data using violin plots was added to the main manuscript. - Some of the samples were from cells grown in organotypic culture and some were from monolayer cell lines. This should be clearly spelled out in the figure legends, and preferably on the graphs themselves.
Response: The new figure now clearly shows this information. - The statistical analysis is strange. In most of the figures, the authors seem to be comparing one group of numbers as a block to another group as a block to determine p values rather than individual data points. They also seem to gloss over what seem to be potentially interesting differences between PCs from sample to sample. One would think that the differences between PC levels at different sites is the thing that the authors are actually looking for, but generally those differences are not generally emphasized.
Response: The new figures showing ratios instead of expression values better visually project the meaning that results from the experiments. - Fig 1B: Is this the average of the 5 anatomical sites included in A? Wouldn’t it be better to normalize the values for each anatomical site to furin and then average the sites together? That would represent the difference in overall levels between the enzymes in a site-agnostic way. Response: Figure 1B shows monolayer cultures and panel A the raft cultures. The presentation of this figure was improved to avoid confusion.
- Fig 2B: it should be indicated on the graph or in the legend which cell types are neoplasias and which are carcinomas.
Response: The new figure is better presented with clear labels. - Fig 4B: what are these numbers and where do they come from? The levels of furin tend to be high in all the other data in the paper, but not as high as in this figure.
Response: This figure was removed as it did not contribute to the story line. - Fig 5: the groupings look hand drawn and vary in style from panel to panel.
Response: The style and quality were improved.

Reviewer 2 Report
Comments and Suggestions for Authors.
Author Response
Thank you for your willingness to make a review and sorry for the misunderstanding - Editor
Reviewer 3 Report
Comments and Suggestions for Authors
The manuscript is interesting and contributes to the topic of HPV-dependent neoplasias. However, some aspects should be improved, according to the following observations:
Introduction. Please indicate the meaning of the abbreviations when they appear for the first time in the text. Although some are described in the abstract, I consider that they should be indicated when they appear in the text, regardless of whether they are described in the abstract.
In the graphs and results, it is not clear what the gene expression count values ​​are. It is understood that it is a relative quantification. Relative to what value. Or is it an absolute quantification (copies/ng of cDNA?). It is not at all clear. Please describe in more detail-
Specify the statistical analyses used, what type of analysis was used, with what number of data. Did you use parametric or non-parametric statistics, and on what basis, in case of using few data. I consider that you should write a well-founded section on the type of statistical analysis used.
The effect of the viral genome on total PC gene expression was minimal (Fig-90 ure 4B).: Figure 4B should perhaps be placed before Figure 2, as it is mentioned in the text before Figure 2.
The P values ​​do not seem very logical in Figure 2, as in the bars where a very large difference is seen, the P value is not significant. Therefore, it is important to know the type of central tendency and dispersion measure used in each graph and the statistical test used (in each graph), which should be based on its use in a statistical analysis section.
Figure 4: Total PC gene expression values ​​per group of cell types (Panel 128 B). Panel B is not clearly described, as are the dispersion measures, nor the number of data represented in each bar.
In the background, and especially in the discussions, it should be made clear what is known about the role of FURIN in HPV infections, and what exactly this manuscript contributes with respect to others. There are already precedents of the relevant role of FURIN in HPV infection and carcinogenesis. For example, the following references are not cited:
https://www.mdpi.com/2072-6694/15/19/4878
https://journals.asm.org/doi/10.1128/jvi.00038-16
https://onlinelibrary.wiley.com/doi/full/10.1002/cti2.1073
Author Response
The manuscript is interesting and contributes to the topic of HPV-dependent neoplasias. However, some aspects should be improved, according to the following observations:
Introduction. Please indicate the meaning of the abbreviations when they appear for the first time in the text. Although some are described in the abstract, I consider that they should be indicated when they appear in the text, regardless of whether they are described in the abstract.
Response: Abbreviations were added at the beginning of the paper body.
In the graphs and results, it is not clear what the gene expression count values ​​are. It is understood that it is a relative quantification. Relative to what value. Or is it an absolute quantification (copies/ng of cDNA?). It is not at all clear. Please describe in more detail-
Response: New figures were made showing the ratios relative to the gene expression determined in normal non-HPV infected cells.
Specify the statistical analyses used, what type of analysis was used, with what number of data. Did you use parametric or non-parametric statistics, and on what basis, in case of using few data. I consider that you should write a well-founded section on the type of statistical analysis used.
Response: More information was added in the Materials and Methods section clarifying the use of independent Student t-test (parametric).
The effect of the viral genome on total PC gene expression was minimal (Figure 4B).: Figure 4B should perhaps be placed before Figure 2, as it is mentioned in the text before Figure 2.
Response: New figures were made and ordered in a better logical manner.
The P values ​​do not seem very logical in Figure 2, as in the bars where a very large difference is seen, the P value is not significant. Therefore, it is important to know the type of central tendency and dispersion measure used in each graph and the statistical test used (in each graph), which should be based on its use in a statistical analysis section.
Response: Figure 2 was redone and now is presented as the ratio relative to the expression in non-HPV infected cells, therefore, the use of p-values was not necessary.
Figure 4: Total PC gene expression values ​​per group of cell types (Panel 128 B). Panel B is not clearly described, as are the dispersion measures, nor the number of data represented in each bar.
Response: A new figure was made and that panel removed.
In the background, and especially in the discussions, it should be made clear what is known about the role of FURIN in HPV infections, and what exactly this manuscript contributes with respect to others. There are already precedents of the relevant role of FURIN in HPV infection and carcinogenesis. For example, the following references are not cited:
Response: The introduction and discussion sections were expanded with more information to strengthen the line of thought followed in the paper. These and other references were added.

Round 2
Reviewer 1 Report
Comments and Suggestions for Authors
The authors adequately responded to my concerns.
Reviewer 2 Report
Comments and Suggestions for Authors.
Reviewer 3 Report
Comments and Suggestions for Authors
The suggestions were taken into account. The manuscript can be published.